# Investigating the ferric ion binding site of magnetite biomineralisation protein Mms6

**Andrea E. Rawlings[1], Panah Liravi[2], Sybilla Corbett[2], Alex S. Holehouse[3], Sarah S. Staniland[1]\***

**1** Department of Chemistry, The University of Sheffield, Sheffield, England, United Kingdom, **2** Faculty of Biological Sciences, The University of Leeds, Leeds, England, United Kingdom, **3** Department of Biochemistry and Molecular Biophysics, Washington University School of Medicine, St. Louis, Missouri, United States of America

\* s.s.staniland@sheffield.ac.uk

**Data Availability Statement:** All relevant data are within the manuscript and its Supporting Information files.

## Abstract

The biomineralization protein Mms6 has been shown to be a major player in the formation of magnetic nanoparticles both within the magnetosomes of magnetotactic bacteria and as an additive in synthetic magnetite precipitation assays. Previous studies have highlighted the ferric iron binding capability of the protein and this activity is thought to be crucial to its mineralizing properties. To understand how this protein binds ferric ions we have prepared a series of single amino acid substitutions within the C-terminal binding region of Mms6 and have used a ferric binding assay to probe the binding site at the level of individual residues which has pinpointed the key residues of E44, E50 and R55 involved in Mms6 ferric binding. No aspartic residues bound ferric ions. A nanoplasmonic sensing experiment was used to investigate the unstable EER44, 50,55AAA triple mutant in comparison to native Mms6. This suggests a difference in interaction with iron ions between the two and potential changes to the surface precipitation of iron oxide when the pH is increased. All-atom simulations suggest that disruptive mutations do not fundamentally alter the conformational preferences of the ferric binding region. Instead, disruption of these residues appears to impede a sequence-specific motif in the C-terminus critical to ferric ion binding.

## Introduction

Biomineralization is the process of forming inorganic minerals under biological control and encompasses the production of calcium carbonates, calcium phosphates, and silicates amongst others [1–4]. One example is magnetic nanoparticles (MNP) synthesised by magnetotactic bacteria [5]. This diverse range of aquatic bacteria share the capability to synthesise single crystals of the iron oxide magnetite inside dedicated organelles termed magnetosomes [6–8], Fig 1.

The magnetosome comprises a lipid bilayer vesicle that surround the MNP as shown in Fig 1, and harbours a large number of specialised proteins. These function to load the vesicle with soluble iron ions, to nucleate the growth of the crystal and ensure adequate maturation of the particle to produce not only the appropriate iron oxide, magnetite, but also with a species specific size and morphology [8–10]. Four magnetosome membrane specific (Mms) proteins were identified as being tightly associated with the magnetite nanoparticles in

**Funding:** A. Rawlings and S Staniland were funded by Biotechnology and Biological Sciences Research Council (Grant No. BB/H005412/1 & 2) https://bbsrc.ukri.org. Panah Liravi & Sybilla Corbett were project students at the time. Sybilla was then part of a EPSRC doctiral training centre, https://epsrc.ukri.org. Alex Holehouse was supported by the Human Frontiers Science Program (grant RGP0034/2017) https://www.hfsp.org. The funders had no role in study design, data collection and analysis, decision to publish, or preparation of the manuscript.

**Competing interests:** The authors have declared that no competing interests exist.

*Magnetospirillum magneticum* AMB-1 [11]. Designated Mms5, Mms6, Mms7, and Mms13 these proteins, due to their close interaction with the nanoparticles, were considered likely candidates for controlling particle formation *in vivo*. An Mms6 deletion mutant displays an irregularly sized and misshapen nanoparticle phenotype [12], indicating a key role in the formation of the nanoparticles, although other deletions using different methods show less effect, demonstrating it is likely Mms6 works in concert with other Mms proteins *in vivo* [13]. Proteins such as MmsF have recently been shown to also have a significant role in this process and are closely located in the Mms6 gene cluster [13, 14]. The addition of purified Mms6 to simple synthetic magnetite precipitation reactions improves the homogeneity of the resulting nanoparticles [11, 15, 16]; a prerequisite for use in biomedicine [17].

Mms6 is a small membrane interacting protein with a low complexity hydrophobic *N*-terminal region, a predicted central transmembrane helix, and a hydrophilic, acid rich C-terminal region, predicted to present on the interior of the magnetosome lumen, so will be exposed to the forming magnetite [11]. Unlike the majority of bacterial proteins, with the exception of the transmembrane helix, the remainder of the protein is predicted to be intrinsically disordered, meaning it does not adopt a stable three-dimensional structure, but instead exists in an ensemble of different conformations [18] (S1). This prediction is supported by previous NMR experiments that showed random-coil chemical shifts in the C-terminal region, but there is also some evidence for the formation of transient helicity, a common structural feature utilized by intrinsically disordered regions when binding [10, 19].

Although predicted to be an integral membrane protein, it is possible to purify overexpressed Mms6 from *E. coli* using protein refolding strategies [11, 15] which result in soluble Mms6 oligomers. Previous studies show this amphiphilic protein self-assembles into 10 nm sized micelles in aqueous solution [20, 21]. With a hydrophobic core, it displays the hydrophilic C-terminus on the micellar surface and demonstrates the ability to bind ferric [11, 20, 21] and ferrous [19] ions with high affinity, therefore implicating the C-terminal acid rich region as the site of iron binding [19–24]. This activity, coupled with the self-assembly (also

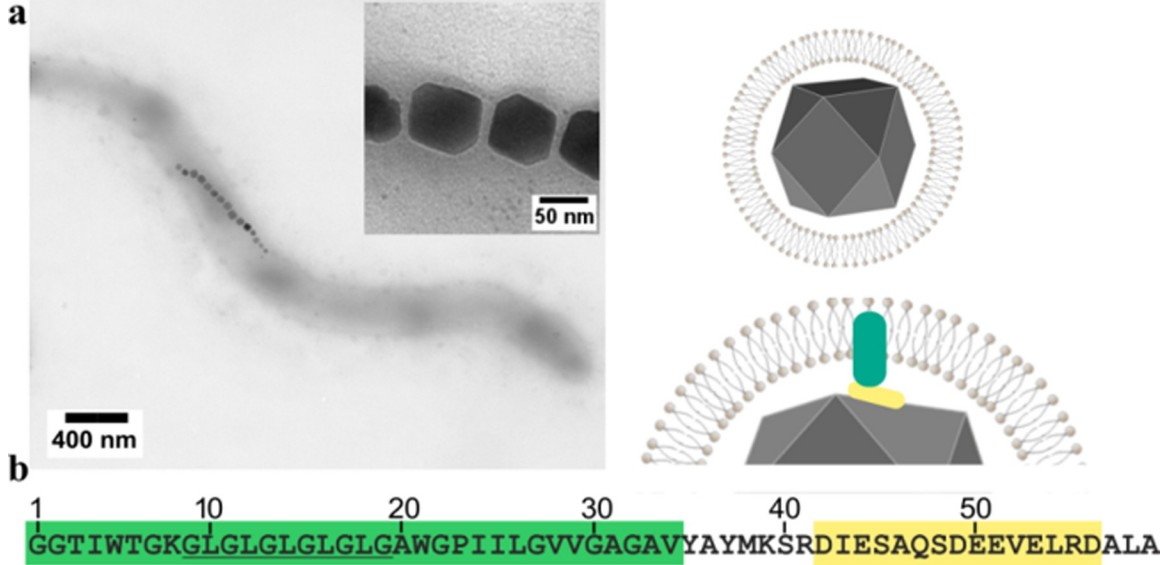

**Fig 1.** (a) Transmission electron microscopy image of *Magnetospirillum magneticum* AMB-1, with schematic of the magnetosome shown. (b) Sequence of Mms6 alongside residue numbering used in this paper. The hydrophobic part in green, Gly-Leu repeat motif underlined and the acidic C-terminal region in yellow.

thought to aggregate within the native magnetosome membrane [25]) may act to display a negatively charged iron binding surface to produce a locally high iron concentration that is predicted to be the magnetite nucleation site. This process has been observed during *in situ* transmission electron microscopy analysis of Mms6 iron complexes [23]. Likely due to a combination of the protein's intrinsically disordered nature coupled with the presence of a transmembrane helix, crystallisation has not been possible to provide full structural characterisation. Recently, NMR was used to obtain amino acid specific structural information about the acidic C-terminal region when bound to various metal ions (ferric, ferrous, calcium and zinc) [19]. A significant structural change was produced in all four acidic residues of the DEEVE (residues 49–53) motif when this peptide was introduced to ferrous ions, with computational modelling highlighting interactions with the E51 and the carbonyl backbone between E50/E51, but surprisingly, no significant conformational change was seen on binding ferric ions [19]. *In vivo* studies of mutations within the C-terminal region of Mms6 have also implicated this DEEVE motif as important for morphology control of the resulting magnetosome particles [26]. Therefore, while there is now some insight into the binding site of ferrous ions, the precise residues involved in ferric binding have not yet been determined.

## Materials and methods

### Strains and plasmids

For cloning and mutagenesis *E.coli* strain XL10 Gold (Agilent) were used. For protein expression a pET28 (Merck) based vector was used which encodes an N-terminal hexa-histidine tag and SUMO fusion (McPherson group Univ. Leeds, UK) and bears the kanamycin resistance gene. For GFP, a vector with a pTac promoter and carbenicillin resistance was used which encodes an N-terminal poly-histidine tag (Baldwin Group, Univ. of Leeds, UK). Mms6 and Mms6MM were produced from carbenicillin resistant pPRIBA1 vectors (IBA). Mms6, Mms6MM were produced in BL21 (DE3) RIL (Agilent) cells, and BL21 (DE3) Star for the SUMO-Mms6 and GFP constructs.

### Media

Two types of media were used. Luria Broth (LB) was used for growing plasmid preparation cultures (0.5% Yeast Extract, 1% Tryptone, 1% NaCl, pH 7.2). For protein production a commercially available autoinducing [27] Super Broth growth media was used which was supplemented with trace elements (Formedium) to support high density culture conditions.

### Gene synthesis and cloning

**For the SUMO-Mms6 constructs.** The mature *mms6* gene sequence was codon optimised for high level *E. coli* production. Silent mutations were introduced to use the codons commonly associated with highly expressed and constitutively expressed genes during exponential growth in *E. coli*. Regions of high GC content and repetitive sequence were also silently mutated to make them more amenable to subsequent mutagenesis reactions. The optimised gene sequence was synthesised by Genscript and also introduced a sequence encoding a Strep II tag and a TEV cleavage site to the 5' termini along with a *Bsa*I site and a 3' *Hind*III site. The sequence was supplied in a pUC57 shuttle vector. The synthesised fragment was excised from the vector using these unique restriction sites and ligated into the similarly cut pET28SUMO vector which encodes an N-terminal yeast SUMO fusion with a hexa-histidine tag before introduction into XL10 Gold (Agilent) cells with transformants selected by resistance to kanamycin. Successful incorporation of the *mms6* sequence was determined by DNA sequencing.

**Protein sequence of SUMO-Mms6 construct.** `MGSSHHHHHHGSGLVPRGSASMSDSEVN`
`QEAKPEVKPEVKPETHINLKVSDGSSEIFFKIKKTTPLRRLMEAFAKRQGKEMDSLRFLYDG`
`IRIQADQTPEDLDMEDNDIIEAHREQIGGCWSHPQFEKENLYFQGASGGTIWTGKGLGLGLG`
`LGLGAWGPIILGVVGAGAVYAYMKSRDIESAQSDEEVELRDALA`

**Mms6 codon optimised DNA sequence used in SUMO construct.** `GGTGGTACCATCTG`
`GACCGGTAAAGGCCTGGGTCTGGGCCTGGGTCTGGGCCTGGGTGCTTGGGGTCCGATCATCC`
`TGGGTGTTGTTGGTGCGGGTGCGGTTTACGCGTACATGAAATCTCGTGACATCGAATCTGCG`
`CAGTCCGACGAAGAAGTTGAACTGCGTGACGCGCTGGCG`

**For the cysteine-Mms6 constructs.** Synthetic gene strings of the sequences shown below were purchased from Life Technologies. The gene strings were digested with *BsaI* (New England Biolabs) and ligated into a similarly processed pPR-IBA1 vector (IBA). The constructs feature a stop codon to prevent read through of the encoded C-terminal StrepII tag on the vector. Plasmids harbouring the sequences were identified by DNA sequencing. The sequence of the constructs is shown below.

**Cysteine Mms6 construct.**

- `ATGTGTGGTAGCCATCATCATCACCATCATGGTAGCGGTGGCACCATTTGGACCGGTAAAG`
  `GTCTGGGCCTGGGACTGGGGCTGGGCCTGGGTGCATGGGGTCCGATTATTCTGGGTGTTGT`
  `TGGTGCCGGTGCAGTTTATGCATATATGAAAAGCCGTGATATTGAGAGCGCACAGAGTGAT`
  `GAAGAGGTTGAACTGCGTGATGCACTGGCATAACCTGCAGGCTAAAGCGCTGAGACCTACC`
  `AT`

- `MCGSHHHHHHGSGGTIWTGKGLGLGLGLGLGAWGPIILGVVGAGAVYAYMKSRDIESAQSD`
  `EEVELRDALA`

**Cysteine Mms6 multi-mutant (Mms6MM) construct.**

- `ATGTGTGGTAGCCATCATCATCACCATCATGGTAGCGGTGGCACCATTTGGACCGGTAAAG`
  `GTCTGGGCCTGGGACTGGGGCTGGGCCTGGGTGCATGGGGTCCGATTATTCTGGGTGTTGT`
  `TGGTGCCGGTGCAGTTTATGCATATATGAAAAGCCGTGATATTGCAAGCGCACAGAGTGAT`
  `GCCGAAGTTGAACTGGCAGATGCACTGGCATAACCTGCAGGCTAAAGCGCTGAGACCTACC`
  `AT`

- `MCGSHHHHHHGSGGTIWTGKGLGLGLGLGLGAWGPIILGVVGAGAVYAYMKSRDIASAQSD`
  `AEVELADALA`

**Protein sequence of His$_8$-GFP.**

- `MGSHHHHHHHHGSTENLYFQGPRMSKGEELFTGVVPILVELDGDVNGHKFSVSGEGEGDAT`
  `YGKLTLKFICTTGKLPVPWPTLVTTFSYGVQCFSRYPDHMKRHDFFKSAMPEGYVQERTIS`
  `FKDDGNYKTRAEVKFEGDTLVNRIELKGIDFKEDGNILGHKLEYNYNSHNVYITADKQKNG`
  `IKANFKIRHNIEDGSVQLADHYQQNTPIGDGPVLLPDNHYLSTQSALSKDPNEKRDHMVLL`
  `EFVTAAGITHGMDELYK`

## Site directed mutagenesis

Mutations were introduced to the pET28SUMOmms6 by site directed mutagenesis (S2). Overlapping primers carrying the desired mismatch were synthesised (Sigma) and used to amplify the vector using KOD hot start DNA polymerase (Merck). After 16 rounds of amplification the reaction was supplemented with *DpnI* restriction endonuclease to digest the methylated template vector before introduction into XL10 Gold cells (Agilent). Plasmid DNA was isolated from a number of colonies generated by each mutagenesis reaction. Successful mutants were identified by DNA sequencing.

## Protein production

The plasmid encoding the desired construct, was introduced to the relevant *E. coli* expression strain (see Strains and Plasmids section). A single colony from each transformation reaction was cultured in LB for 8 hours at 37˚C with shaking in the presence of the required antibiotics. 1 ml of this pre-culture was added to 400 ml of Super broth autoinduction media (Formedium) supplemented with antibiotics. The cultures were grown in baffled 2 L Erlenmeyer flasks for 24 hours at 37˚C with vigorous shaking to ensure sufficient aeration. The cells were subsequently harvested by centrifugation before storage at -80˚C.

## Protein purification

**For SUMO-Mms6, variants and GFP.**   The cell pellets were resuspended in Buffer A (25 mM Tris pH 7.4, 100 mM NaCl) to give a 20% w/v ratio and protease inhibitor cocktail III (Merck) was added. The cells were homogenised fully before lysing with sonication (5 x 30 second bursts) in an ice bath. Insoluble material was removed by centrifugation using a JA25:50 rotor (Beckman Coulter) at 12, 000 rpm for 45 minutes. The pellet was discarded and the supernatant retained. The soluble protein fraction was applied to a 1 ml HiTrap Nickel charged column using an Akta Explorer (GE Lifesciences). After sample loading the column was washed with buffer A for 10 ml to remove unbound proteins, followed a wash with Buffer A containing 50 mM imidazole to remove weakly bound proteins. Finally, the hexahistidine tagged protein was eluted from the column by application of Buffer A with 300 mM imidazole. The absorbance at 280 nm was monitored to determine the peak fractions. These were pooled and supplemented with 20 mM EDTA before dialysing against Buffer A, and then finally against 20 mM Tris treated with Chelex. This acts to reduce trace metal ion contamination of the sample proteins. (Yield S3, SDS PAGE S4)

To check the oligomeric state of the protein, Sumo-Mms6 was subjected to gel filtration using a Superdex S200 10/300 column (GE Lifesciences). The column was equilibrated in 20 mM Tris pH 7.4 to match the buffer of the protein sample. A calibration was carried out using the Gel Filtration Marker Kit (Merck), and the molecular weight of the SUMO-Mms6 peaks extrapolated from comparison to these standards (S5).

**For the Mms6 and cysteine Mms6.**   The cell pellet was resuspended, homogenised, lysed and cleared exactly as described for the SUMO-Mms6. After centrifugation the pellet was washed with more Buffer A, and centrifuged again, and the supernatants discarded. The pellet was resuspended in 8M Guanidine hydrochloride with 300 mM NaCl, and 25 mM Tris pH 7.4, and 5 mM DTT (denaturation buffer). After 1 hour to allow denaturation of the proteins, the suspension was subjected to a further round of centrifugation to remove insoluble material. The supernatant containing Mms6 was mixed with 0.5 ml of Amintra Nickel-NTA resin (Expedeon, UK) for 30 minutes with gentle agitation. The supernatant was removed, and the resin transferred to a gravity flow column and washed with further denaturation buffer supplemented with 30 mM imidazole pH 7.4. Finally the Mms6 was eluted with denaturation buffer containing 300 mM imidazole in 0.5 ml fractions. The fractions containing Mms6 (estimated from UV 280 nm absorbance, volume approximately 2 ml) were pooled and added dropwise to 200 ml of vigorously stirred 500 mM NaCl, 25 mM Tris, pH 7.4, and 1 mM DTT to induce rapid refolding. The protein was then concentrated using a 10 kDa molecular weight cut off Amicon centrifugal concentrator (Merck Millipore). Once concentrated, the protein was dialysed using a 3.5 kDa molecular weight cut off dialysis tubing against 500 mM NaCl (SDS PAGE S6). The protein was stored at -80˚C.

## Iron binding assay

The absorbance of the proteins was measured at 280 nm and the protein concentration was estimated based on their theoretical extinction coefficient, and normalised to 1 mg/ml. The proteins were supplemented with freshly prepared ferric citrate to produce a final iron concentration of 1 mM, and a final protein concentration of 42 μM (63 μl protein and 7 μl ferric citrate). After one hour incubation at room temperature, shielded from light to prevent reduction of the ferric ions, the protein-iron samples were desalted using 7 kDa molecular weight cut off desalting spin columns (Zeba columns, Thermo). Each sample was desalted twice to ensure thorough removal of the unbound iron ions.

**For the luminescence based detection.** 8 μl (0.34 nmol) of each of the desalted protein samples was added separately to 16 μl of 8 M Urea in a 96 well luminescence assay plate and allowed to denature for 30 minutes at room temperature. 40 μl of freshly prepared luminescence reagent was added (500 mM $Na_2CO_3$, 11 mM luminol, and 230 mM $H_2O_2$)[28] to each well and the plate transferred promptly to a Fluostar Optima plate reader (BMG labtech) for luminescence measurements. The assay was repeated three times and the values averaged.

## Nanoplasmonic sensing

An Xnano instrument (Insplorion) was used with supplied bare gold glass sensor chips which were cleaned for 10 minutes using an ozone cleaning device. Dark and bright field spectra were recorded. The system was washed with ultrapure water using a peristaltic pump set to 50 μl per minute. Temperature was maintained at 25°C. A spectrum relating to the Au plasmon peak was recorded each second while the system was operating. After obtaining a stable plasmon peak in water, the system was transferred to buffer (25 mM Tris pH 7.4, 500 mM NaCl) until the plasmon peak stabilized again. At this point protein was applied at the same flow rate as before, at a concentration of 0.3 mg/ml. After a few minutes a stable plasmon peak was obtained. The system was then returned to buffer and finally water. A 50 mM iron sulphate solution was prepared. This solution comprised a 2:1 ratio of Fe(III) to Fe(II), the stoichiometric ratio of magnetite, dissolved in nitrogen sparged ultrapure water. The solution was introduced to the Xnano at the same flow rate as before until a stable plasmon peak was observed. The iron inlet was then returned to water and the flow maintained.

To aid the analysis, the transition times at which the solutions were switched was recorded. This is the buffer to protein transition, the water to iron solution transition, and the iron solution to water transition.

A 400 second window surrounding each transition point was created which encompassed 100 seconds before, and 300 seconds after the switch point. The stable wavelength prior to the solution switch (first 100 seconds) was taken as a stable baseline and subtracted from all the wavelengths in this window to produce a change in wavelength value ($\Delta\lambda$) at each one second interval over this timeframe. This allowed the different experiments (Mms6, Mms6MM and protein free) to be overlaid.

## Monte Carlo simulations

All-atom Monte Carlo simulations were run using the CAMPARI Monte Carlo simulation engine and the ABSINTH implicit solvent paradigm using the abs_3.2_opls.prm parameters and the ion parameters of Mao et al [29, 30]. Simulations were run at 298 K with 20 mM NaCl. For each construct, 20 independent simulations were run for $30 \times 10^6$ production steps with $2 \times 10^6$ equilibration steps in a simulation droplet with a radius of 56 Å. Configurations were sampled every $20 \times 10^3$ steps during production simulations, such that each independent simulation provided 1500 decorrelated configurations. In total, each construct provides an

ensemble of $30 \times 10^3$ independent conformations. Simulations were analysed using MDTraj and CAMPARITraj (https://github.com/holehouse-lab/camparitraj/) [31].

## Results and discussion

### Ferric iron binding investigation of Mms6 and mutants

A range of soluble SUMO-tagged Mms6 mutants were designed to probe the contribution of specific residues to the overall ferric binding activity of the protein. Amino acids with acidic side chains (glutamic and aspartic acid) were targeted as these will typically be negatively charged at neutral to basic pH, which is required for magnetite formation and stability, and would therefore be able to coordinate positively charged iron ions. Other commonly associated iron binding residues such as histidine, cysteine, and tyrosine are not present within the acidic region of Mms6. Aspartic acids (D) at positions 42, 49 and 56, and Glutamic acids (E) at positions 44, 50, 51 and 53 were all individually substituted to alanine giving mutants designated 'D42A' for instance. Additionally, a double mutant EE50AA was created by mutating two adjacent glutamic acids to alanine residues. The arginine residue located at position 55 is conspicuous as being a lone residue of the opposing charge state within this highly acidic region and its role was also investigated by substitution to alanine.

In order to probe how Mms6 iron binding was affected by the introduced substitutions we used a radiolabel-free iron binding assay based on a system developed by Hogbom *et al* [28] (Fig 2A and 2B), which uses Fenton chemistry and luminol to generate a luminescent signal, demonstrating an effective quick screening methodology for our range of Mms6 mutants. GFP was also produced to act as a negative control protein which does not interact with iron ions but has a poly-histidine tag as Mms6.

Mms6 and the Mms6 variants were expressed in *E. coli* as a fusion to an N-terminal His6-SUMO tag; the His6 tag to enable purification by nickel affinity chromatography, and the SUMO to maintain protein solubility. It should be noted that these tags are likely to affect/

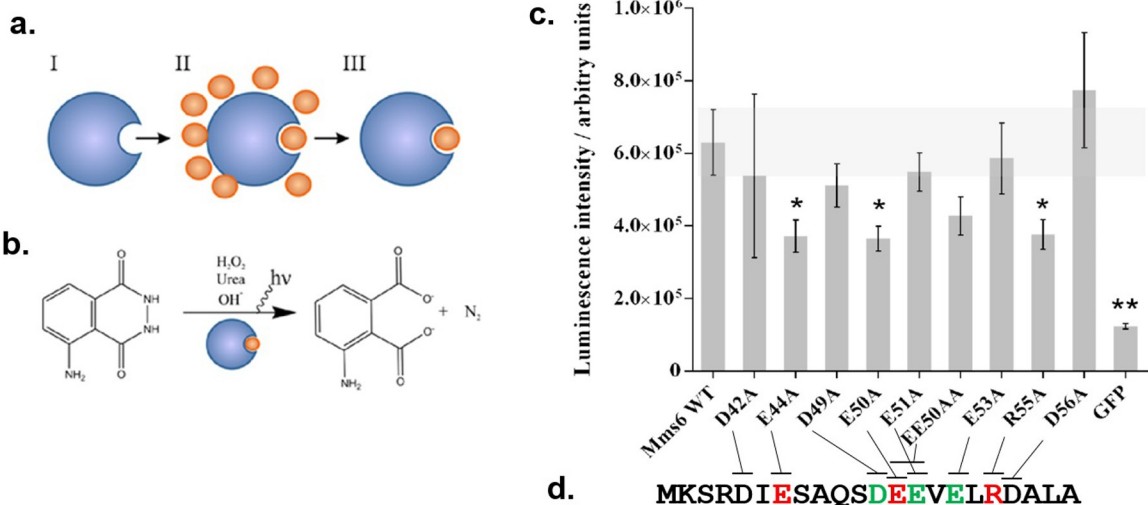

**Fig 2.** (a) Overall assay scheme. (I) De-metallized Mms6 is incubated with excess ferric citrate (II), before the excess is removed by a desalting procedure (III). (b) The iron bound protein from (III) is mixed with luminol, hydrogen peroxide, urea and hydroxide which releases iron from the protein and causes subsequent reaction of the luminol and emission of light. (c) Averaged luminescence intensity obtained from three replicates of each protein sample with error bars indicated (standard error of mean). Statistically significant values are identified by an asterisk using 1 way Anova (p = 0.05). (d) C-terminal sequence of Mms6 with the sites that are critical to ferric binding highlighted in red, and the sites found critical in ferrous binding in green (from a previous study [19]).

inhibit Mms6 self-assembly. Size exclusion chromatography analysis of SUMO-Mms6 reveals two species. The major species has a retention volume equivalent to monomeric SUMO-Mms6, and the minor species elutes at a volume consistent with dimeric SUMO-Mms6. Similar results are observed in SDS-PAGE (S4, S5). The proteins were de-metallized to reduce background luminescence by dialysis against EDTA then Chelex treated ultrapure water. Protein concentrations were normalized based on their absorbance at 280 nm. Each protein was incubated with ferric citrate before being rapidly desalted (Zeba desalting spin columns, Thermo-Pierce) to remove unbound metal ions. Proteins were then denatured to release bound metal ions and luminol based detection reagent applied to each sample before measuring the luminescence intensity.

As expected, the GFP negative control gave the lowest luminescent signal indicating that the desalting procedure sufficiently removed most unbound ferric ions from the protein incubation step, generating a negligible background luminescence. Mms6 gave a significantly higher value indicating that ferric ions bound to and were retained by the protein (Fig 2C). To sequester iron from the citrate complex Mms6 must have a higher binding affinity than citrate, which is consistent with previous Mms6 studies [20].

Using this assay several of the Mms6 variants gave luminescent intensities close to that of wild type Mms6, suggesting that the substitution of those particular residues to alanine did not significantly impair the iron binding capability on their own. These residues include glutamate 51 and 53, and all the aspartate residues in the C-terminus: 42, 49 and 56 (Fig 2D).

Four of the variants E44A, E50A, EE50AA and R55A gave a luminescence intensity below wild type Mms6, indicating impaired ferric ion binding. However, neighbouring E51 does not appear to significantly affect ferric binding in this study either alone or when combined in the double substitution, suggesting the effect of the double mutant is solely due to the effect of E50A.

Our studies also implicate R55 in ferric binding. Although unlikely to interact directly with the ferric ions, we speculate that as one of the few basic residues in this region of the protein it may play a structural role by electrostatically mediating the geometry of the binding site, or in the coordinating of a multimeric Mms6 binding complex.

## Nanoplasmonic surface investigation of Mms6 triple mutant

Overall, the luminol screening assay has shown E44, E50 and R55 to be amino acid sites implicated in ferric binding. We produced an Mms6 triple mutant comprising alanine substitutions of the three key residues identified EER44,50,55AAA Mms6 (designated Mms6MM for Mms6 multi-mutant) to investigate the cumulative effect of all three mutations. It should also be noted that the deletion of the more basic arginine with the two glutamic acids, goes some way to ensuring the overall charge of this variant is only + 1 different, so the same as the other acidic single mutants. Unfortunately, the aqueous stability of this variant was extremely low, making purification challenging, and characterisation within the ferric iron binding assay impossible. However, we were able to use a surface based nanoplasmonic sensing experiment (using the Xnano (Insplorion) [32]) to probe Mms6MM and compare it to native Mms6 in terms of their interaction with iron ions and their ability to promote/ impede the precipitation of iron-oxide *in vitro*. Previous research has demonstrated how Mms6 is able to promote the formation of magnetite when assembled on a gold surface as this surface assembly better mimics the native environment on the membrane within the magnetosome [25, 33, 34]. The absence of the SUMO tag thus ensures any self-assembly is not restricted. Using bare gold sensors allowed us to immobilise Mms6 and Mms6MM via a single cysteine residue incorporated in the N-terminal tag (as we have shown previously [25, 33, 34]). The change in wavelength of

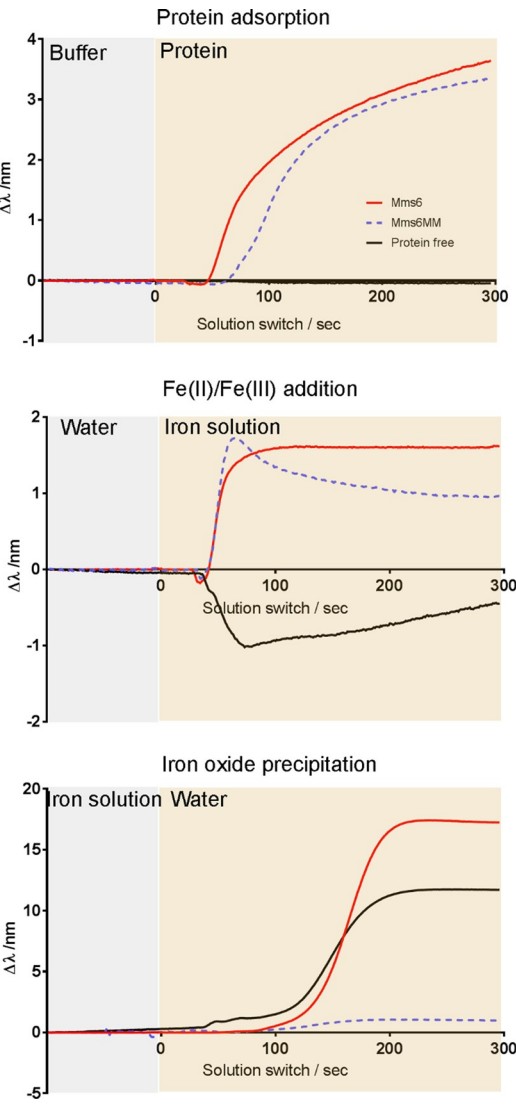

**Fig 3.** Nanoplasmonic sensing of: a) protein adsorption, b) iron binding, and c) iron oxide precipitation. Red line is Mms6, blue dotted line is the Mms6MM, and the black line is the protein free experiment. Each panel depicts the transition between two solutions in the experiment, with time zero being the transition point in each case The change in the wavelength of the plasmon peak for each experiment is shown.

the plasmon peak of the gold sensor was monitored whilst immobilising the protein, applying a mixed valence iron solution, and washing with ultrapure water (Fig 3). Shifts in wavelength can be interpreted as changes in the species bound to the sensor [32]. It appears that both proteins attached in a similar manner to the gold surface, as evidenced by comparable shifts in the plasmon peak position showing similar levels of surface coverage to near saturation after approximately 5 minutes (Fig 3A). In the case of the protein free experiment, buffer alone was applied to the sensor in place of the protein. Any excess protein in the sensor chamber was removed by application of ultrapure water to the sensor surface before application of an iron solution. This resulted in a small shift in wavelength for both protein experiments compared to the protein free sensor. We interpret this as either an interaction between the iron ions and the protein, a rearrangement of the protein structure upon application of the iron, or a combination of the two factors. This latter interpretation is consistent with previous reports of

Mms6 iron binding and structural rearrangement [19–24]. It is noteworthy that the signal for Mms6MM decays over time. This suggests that the strength of binding to iron ions in Mms6MM is weaker than the native Mms6 interaction, and points to differences in behavior between the two proteins under the same conditions. It is predicted the binding in Mms6MM is of ferrous ions only (due to ferric binding inhibition), or weaker unspecific general binding (Fig 3B).

Following application of the iron ions, the system was returned to ultrapure water. In the Mms6 experiment this resulted in a large shift in wavelength, much more significant than that observed for application of either protein or iron ions. A similar response was observed in the protein free condition. Such a large shift in the position of the plasmon peak indicates a highly significant change in the environment close to the sensor surface. A potential explanation is precipitation of iron oxide due to the increase of pH from the acidic iron solution to the pH of neutral ultrapure water. Ferric iron oxides can precipitate at $pH \leq 6$ whereas, ferrous iron is more soluble with ferrous iron oxides only precipitating at $pH > 6$ [19, 35]. Interestingly, the shift was not observed in the Mms6MM experiment (Fig 3C). These results show Mms6 and Mms6MM coated sensors behave differently when the pH of the iron solution is increased to return to ultrapure water, whereby Mms6MM appears to inhibit the significant environmental change observed in the Mms6 and protein free experiments. At this pH ferric ions should precipitate as the oxide but ferrous ion should stay soluble. If, as suggested by the iron binding assay Mms6MM ferric iron binding is inhibited, then this would block the formation of an iron oxide precipitate on the surface of the protein on the sensor at this pH. However, we cannot exclude the possibility that the substitution of the three residues in Mms6MM has altered the protein structure, leading to a conformationally inactive form of Mms6 which may just block the sensor surface.

## Monte Carlo simulation of Mms6 and mutants

The mutations that reduce ferric binding may exert their influence through two non-mutually exclusive mechanisms: they may disrupt specific amino acids that engage in direct interaction with the ferric ion (disrupting *inter*-molecular interactions), or they may lead to significant changes in the underlying structure and conformational preferences of the C-terminal region (disrupting *intra*-molecular interactions) indirectly rendering this region less binding competent. To differentiate between these two scenarios, we performed extensive all-atom simulations of the C-terminal 21 residues for the WT sequence and all mutants. Given this region is predicted to be intrinsically disordered (S1), simulations were performed with the ABSINTH implicit solvent paradigm and CAMPARI Monte Carlo simulation engine [30]. The combination of ABSINTH and CAMPARI has been used extensively to characterize atomistic ensembles of intrinsically disordered regions and proteins [36–39].

All simulations led to expanded and conformationally heterogeneous ensembles, consistent with this region being intrinsically disordered (S1, S7, S8 and movie SM1). In agreement with previous predictions, simulations of all the constructs consistently identified transient helicity (10–15%) in the C-terminal region of the peptide (Fig 4B). Bioinformatic predictions using MoRFchibi SYSTEM identified two putative molecular recognition features (MORFs) in this region (Fig 4A) hinting at possible binding sites [40]. With these predictions in mind, it might be expected that mutations inside these MORFs would have a greater impact on the conformational behaviour of the peptide than mutations in surrounding residues. To assess this, we quantified the global and local conformational preference for each construct compared with wildtype, as well as the conformational heterogeneity for each ensemble (Fig 4, see methods for details). We also considered if mutations that significantly attenuated iron binding (E44A,

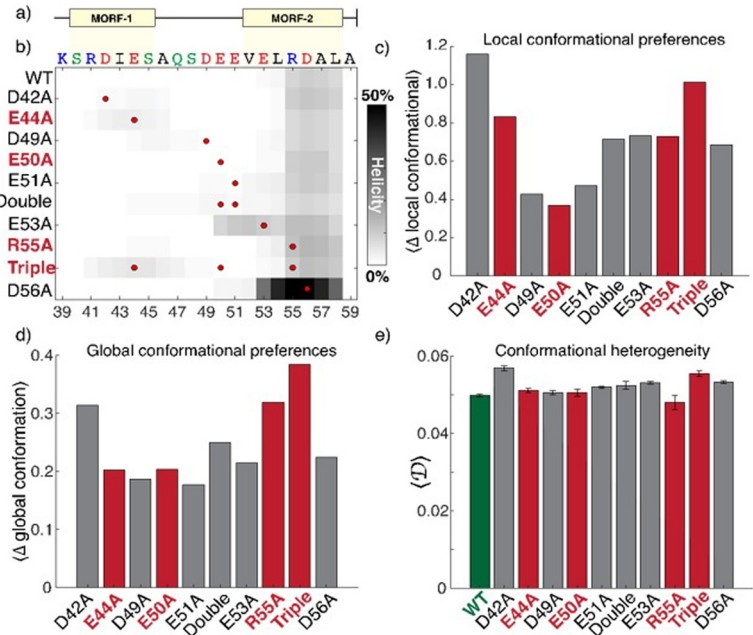

**Fig 4. All-atom simulations of C-terminal residues 38–59.** a) Bioinformatics analysis of the primary sequence identified two putative molecular recognition features (MORFs) in the C-terminal region. **b)** All mutations except E53A and D56A had a minimal impact on residual helicity. The mutation position is shown as a red circle, darker shades represent enhanced helicity. **c)** Local conformational differences are calculated as the deviation from wildtype in terms of all pairwise intramolecular distances (see also S2 Fig). The mutants that show the smallest (E50A) and largest (triple) deviation from WT both significantly reduce iron binding, suggesting local interactions are not a useful metric for assessing the determinants of iron binding. **d)** Global conformational preferences are calculated in terms of the deviation from ensemble average shape (asphericity) and size (radius of gyration) 2D distributions (see also S3 Fig). Global conformational behaviour is again not predicting of iron binding with no discernible correlations identified. **e)** Global heterogeneity quantifies the extent of conformational disorder in terms of the distribution of structurally dissimilar conformations [49]. Larger values indicate higher heterogeneity. There is no correlation between heterogeneity and iron binding ability.

E50A, R55A, or the triple mutant) seemed more similar to one another in terms of their impact on the conformational behaviour when compared to those that had no effect.

Irrespective of the impact on ferric binding, all the mutations examined led to relatively minor and similar changes in the conformational properties of the ensemble (Figs 4B–4E and S7 and S8). More importantly, we did not identify any trends between the extent of global or local conformational change and the impact on iron binding. For example, the mutants with the smallest and greatest deviation from wildtype conformational behaviour (Fig 4C, see also S8 Fig) were E50A and the triple mutant, respectively, both of which also reduced ferric binding. This suggests that across the mutants tested, the intrinsic conformational ensembles are not predictive of the ferric binding capacity of a given mutant. Given the challenges associated with parameterizing fixed-charge models for metal ions and the absence of current support in the ABSINTH implicit solvent model we did not pursue simulations that included ferric ions.

These results support a model in which the C-terminal region of Mms6 acts as a conformationally malleable binding interface, whereby specific residues engage in direct interactions with iron. Considering many disordered regions undergo helix formation upon binding to a cognate partner [41, 42] we were surprised to notice that changes to intrinsic helicity appeared to have a minimal impact on binding. For example, the neutral mutation D56A led to a substantial enhancement in C-terminal helicity, while another neutral mutation (E53A) extended the

helix towards the N-terminus of the peptide. Given prior NMR results did not detect any increase in helicity upon binding to distinct metals, these results agree with a model in which binding is likely mediated by binding interface that lacks any strong structural biases, in contrast to the folding-upon-binding mechanism often associated with disordered regions [43]. These results are reminiscent of examples in which highly charged intrinsically disordered regions bind without the acquisition of structure in polyelectrolyte complexes, although the mechanism invoked here may be better described in terms of counterion condensation, as has been explored extensively in the context of nucleic-acid:cation interactions [44–46]. Despite this, apparent net charge appears to be a poor predictor of the impact (or lack thereof) of mutations. One explanation for this may reflect the fact that the local electrostatic environment may shift the pKa values of negatively charged residues such that, even at a pH of 7.4 in the absence of ferric iron, some of the ostensibly charged residues may be neutralized due to upshifted pKa values [47, 48].

## Conclusion

In conclusion, the development of a convenient luminescence based binding assay has allowed for the rapid parallel screening of a range of amino acid substitutions of Mms6 for binding to ferric ions with a number of key residues now identified. Namely, the three mutants E44A, E50A and R55A showed significantly reduced ferric binding, revealing E44, E50 & R55 are important for ferric ion binding. This is the first time residue specific ferric binding sites have been identified in Mms6.

The cumulative effect of all three significant mutations were explored in a surface-bound nanoplasmonic assay and showed distinct differences between the binding and precipitation of iron-oxide on the surface of each protein. Mms6MM showed reduced binding to iron ions and did not show a dramatic shift when the pH was raised when compared to native Mms6 on the surface that did. We suggest the reduced binding of Mms6MM is due to the inhibition of ferric ion binding and thus only ferrous ions are bound in this experiement. Ferrous ions will not precipitate to the oxide when the pH is raised to ultra pure water. In the Mms6 case where both ferric and ferrous ions can bind iron-oxide precipitation is then possible and we suggest is the responsible for the large signal shift.

It is interesting to note the whole DEEVE region is implicated in ferrous binding with particular affinity for E51 [19], while E50 and E44 are involved in ferric binding, suggesting Mms6 C-terminal contains specific ferric and ferrous binding sites which are orthogonal to one another (Fig 2D) and supports the hypothesis that Mms6 is a specific magnetite nucleation protein, functioning to specifically bind ferric and ferrous ions in a specific conformation to favour magnetite formation [24].

All-atom simulations agree with extant sequence-based predictions and NMR results that the C-terminal acidic region is intrinsically disordered. Previous NMR results observed a minimal change in helicity upon metal ion binding, implying binding is not strongly coupled to a specific structural changes [19]. In agreement with this, simulations found that changes to the intrinsic helicity driven by mutations had no obvious relationship to the binding affinity as measured experimentally. The simulations and previous NMR address a similar question in entirely independent manners yet arrive at the same conclusion. These results imply that Mms6 binding is likely mediated through a structurally heterogeneous binding interface than can bind equally well through a number of different conformations. Our results provide an example of how intrinsically disordered regions can be used by bacteria to achieve biological function, further challenging the decaying assumption that bacteria lack biologically important disordered regions.

## Supporting information

**S1 Fig.** Predicted structure of Mms6: a) Predicted disordered regions of full length Mms6. Red indicates disorder. b) Mms6 sequence matched to disordered (yellow) and ordered (green) regions within C-terminal region used in SUMO-Mms6 construct. c) Representative simulations of the disordered C- terminal region (yellow) of Mms6. Residue colouring matches that shown in the sequence in b)
(DOCX)

**S2 Fig. Mutagenic primers for the SUMO-Mms6 construct (full length Mms6): Mutagenic primers used in the experiment.** Black boxes indicate codon changes to alanine from the wild-type Mms6 sequence shown. Asterisks highlight residues in Mms6 which were targeted. This figure depicts just the C-terminal region of Mms6 for clarity. The residue numbering cor-responds to the residue position in the truncated form of Mms6 isolated by Arakaki *et al* (1). See cloning section in Supplementary methods section for full sequences of constructs.
(DOCX)

**S3 Fig. Purified proteins: Yield of purified SUMO-Mms6 and variants.** Yield based on UV absorbance at 280 nm and calculated from the theoretical extinction coefficient. This is the yield after purification and dialysis steps were complete.
(DOCX)

**S4 Fig. Purified proteins: SDS-PAGE analysis of SUMO-Mms6 and variants used in the iron binding study.** Bio-Rad AnyKd gel (Bio-Rad) with InstantBlue staining (Expedeon, UK). M is the molecular weight marker (PageRuler, Thermo Scientific) with MW in kDa indicated. Lanes 1–10 are: GFP and wildtype SUMO-Mms6, D24A, E44A, D49A, E50A, E51A, EE50AA, E53A, and R55A.Theoretical MW is approximately 21.5 kDa. Apparent monomers and dimers are present in each lane.
(DOCX)

**S5 Fig. Purified proteins: Gel filtration analysis of SUMO-Mms6.** Table shows the calibra-tion standards (Gel Filtration Markers Kit, Merck) and retention volumes from a Superdex 200 10/300 analytical gel filtration column. Gel filtration plot shows the absorbance at 280 nm for a sample of SUMO-Mms6 as it emerges from the same column. Two species (1 & 2) are highlighted, and their calculated molecular weights are presented in the Table below.
(DOCX)

**S6 Fig. Purified proteins: SDS-PAGE analysis of Mms6 and Mms6MM.** BisTris RunBlue gel with Instant Blue staining (Expedeon, UK). M is the molecular weight marker (PageRuler, Thermo Scientific) with MW in kDa indicated. Purified Mms6MM and Mms6 are shown.
(DOCX)

**S7 Fig. Computational analysis.** Scaling maps for IDRs in Mms6 and mutants. Cooler colours reflect inter-residue compaction compared to the behavior of an excluded volume model, while warmer colours reflect expansion.
(DOCX)

**S8 Fig. Computational analysis: Conformational distributions from atomistic simulations of the Mms6 and mutants in terms of size and shape (asphericity).** Colours are a probability range from blue (zero probability) to yellow (0.004 probability).
(DOCX)

## Acknowledgments

The authors thank Patrik Bjöörn, Jenny Andersson, and David Johansson at Insplorion, Gothernberg, Sweden for use of the Xnano and general assistance. Thanks must also go to the McPherson group, University of Leeds, UK, for the original SUMO construct. The initial work by Alex Holehouse was performed while he was a postdoctoral fellow in the laboratory of Dr. Rohit V. Pappu at Washington University in St. Louis.

## Author Contributions

**Conceptualization:** Andrea E. Rawlings, Alex S. Holehouse, Sarah S. Staniland.

**Data curation:** Andrea E. Rawlings, Alex S. Holehouse, Sarah S. Staniland.

**Formal analysis:** Andrea E. Rawlings, Panah Liravi, Sybilla Corbett, Alex S. Holehouse, Sarah S. Staniland.

**Funding acquisition:** Sarah S. Staniland.

**Investigation:** Andrea E. Rawlings, Panah Liravi, Sybilla Corbett, Alex S. Holehouse, Sarah S. Staniland.

**Methodology:** Andrea E. Rawlings.

**Project administration:** Sarah S. Staniland.

**Supervision:** Andrea E. Rawlings, Sarah S. Staniland.

**Writing – original draft:** Andrea E. Rawlings, Sarah S. Staniland.

**Writing – review & editing:** Andrea E. Rawlings, Sarah S. Staniland.

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
