## [Decision Letter · Decision Letter 0]

6 Nov 2019

PONE-D-19-25528

Investigating the ferric ion binding site of magnetite biomineralisation protein Mms6

PLOS ONE

Dear Dr. Staniland,

Thank you for submitting your manuscript to PLOS ONE. After careful consideration, we feel that it has merit but does not fully meet PLOS ONE’s publication criteria as it currently stands. Therefore, we invite you to submit a revised version of the manuscript that addresses the points raised during the review process.

ACADEMIC EDITOR: Please try to improve your manuscript according to the criticism of the reviewers.

We would appreciate receiving your revised manuscript by Dec 21 2019 11:59PM. To enhance the reproducibility of your results, we recommend that if applicable you deposit your laboratory protocols in protocols.io, where a protocol can be assigned its own identifier (DOI) such that it can be cited independently in the future. For instructions see: http://journals.plos.org/plosone/s/submission-guidelines#loc-laboratory-protocols

We look forward to receiving your revised manuscript.

Kind regards,

Eugene A. Permyakov, Ph.D., Dr.Sci.

Academic Editor

PLOS ONE

Journal Requirements:

"No".

i) Please complete your Competing Interests on the online submission form to state any Competing Interests. If you have no competing interests, please state "The authors have declared that no competing interests exist.", as detailed online in our guide for authors at http://journals.plos.org/plosone/s/submit-now

ii)  This information should be included in your cover letter; we will change the online submission form on your behalf.

Reviewers' comments:

Reviewer's Responses to Questions

**Comments to the Author**

1. Is the manuscript technically sound, and do the data support the conclusions?

Reviewer #1: Yes

Reviewer #2: No

2. Has the statistical analysis been performed appropriately and rigorously? 

Reviewer #1: Yes

Reviewer #2: No

3. Have the authors made all data underlying the findings in their manuscript fully available?

Reviewer #1: Yes

Reviewer #2: Yes

4. Is the manuscript presented in an intelligible fashion and written in standard English?

Reviewer #1: Yes

Reviewer #2: Yes

5. Review Comments to the Author

Reviewer #1: Re: Investigating the ferric ion binding site of magnetite biomineralisation protein Mms6

The presented work an in-depth investigation on the mode of iron ion binding by a protein aiding magnetite crystals formation in magnetotactic bacteria. The study is well planned and conducted and is consistent with previous works on the same protein. In particular, the authors have identified three amino acid residues involved in Fe3+ binding, and demonstrated that those are different from the residues binding Fe2+ ions.

The article could be accepted for publication as is, however I would comment on a couple of points:

1. The Monte-Carlo simulation would have been more sensible if the 2+/3+ metal ions were included in the structure. Although I understand, there might be some limitations in the force field.

2. On Fig. 3c, dotted line could be rather interpreted as that there’s a metal binding to the mutant, but it doesn’t lead to nucleation (for whatever reasons). I would appreciate authors’ comment on that.

Minor and technical comments:

1. Ln. 67: “…nucleate the formation…” is a tautology.

Reviewer #2: This manuscript from Staniland and co-workers describes studies of iron binding by the Mms6 protein from Magnetospirillum magneticum AMB-1. The approach taken used two assays to assess iron binding capability, which are very different in their means of sensing bound iron. One uses only ferric-citrate as a chelated ligand, while the other uses a combination of ferric/ferrous citrate. One is a measurement with proteins in solution and the other measures binding to proteins on a surface. In solution the SUMO-fused proteins are reported here to be monomers and dimers. The non-fusion protein has been shown as a multimer, and contrary to the SUMO-fused proteins, it has been refolded from a denatured state. Although the authors did not investigate the structure of the protein in this study when it is attached to a surface, it is reasonably likely that its multimeric state remains. The interaction between adjacent C-terminal domains or adsorption to the surface could profoundly influence the structure of an intrinsically disordered polypeptide and alter the means by which the protein binds iron compared with the SUMO-fused monomers. As the authors did not measure the affinity for iron in either assay, they do not evidence to support the assumption that the two C-terminal domains have the same structure or that the same iron-binding configuration of residues is evaluated by each assay. Bringing the results from these two assays together to make a single conclusion regarding the binding site of Mms6 for iron is not justified.

Other aspects of the submission are also of concern. First, the procedure for attaching protein to the surface for nanoplasmonic sensing involved a few minutes of flow until a stable plasmonic signal was reached. The authors state that this short period resulted in saturation of the sensor by both proteins, but provide no evidence to support this statement, which is contrary to published results that show it takes about an hour for the reaction between gold and a thiol group to reach saturation (Bard et al. 2018 Photochemistry and Photobiology 94 (6) 1109-1115). Regarding the results from the plasmonic sensing assay the authors note the slight reduction of signal for Mms6MM over time to level of at about 70% of the wild type protein signal and propose, without experimental support, this lower level of binding is the binding of ferrous ions only, or weaker unspecific general binding. The authors also noted a large wavelength shift when the system was returned to water, which they interpreted as being due to the precipitation of iron oxide on the surface of the gold. The lack of shift for the triple mutant protein was interpreted as being the result of the mutant being unable to bind ferric ions. Again there is no experimental support given to this interpretation. At the very least a protein that does not bind iron should have been used as a control condition. It is also not stated if this assay was performed more than once and how the results varied between tests, if done.

Figure 2 is unnecessary as this information including the chemistry involved is published as a figure in an open-source publication that has been cited by the authors

The manuscript also contains typographical errors that need to be corrected.

6. PLOS authors have the option to publish the peer review history of their article (what does this mean?). If published, this will include your full peer review and any attached files.

Reviewer #1: No

Reviewer #2: No

---

## [Author Response · Author response to Decision Letter 0]

20 Jan 2020

Answer. We have been thorugh the style guide and made teh amednemetns as requested. (note that the formating correction (heading etc) are not in the tracked changes document to aviod confusion. Quick question. It states that funding should not be stated in the acknowledgements. Where should this be stated?

"No".

i) Please complete your Competing Interests on the online submission form to state any Competing Interests. If you have no competing interests, please state "The authors have declared that no competing interests exist.", as detailed online in our guide for authors at http://journals.plos.org/plosone/s/submit-now

Answer I have looked at this page and it is not clear what you wish for me to do. have I done everything correctly? Please be in touch with links to any specific forms if not.

ii) This information should be included in your cover letter; we will change the online submission form on your behalf.

Answer I think it was in the last cover letter.

Reviewers' comments:

The answers to the reviwers commetns are in the cover/rebuttal letter

---

## [Editor Report · Decision Letter 1]

23 Jan 2020

Investigating the ferric ion binding site of magnetite biomineralisation protein Mms6

PONE-D-19-25528R1

Dear Dr. Staniland,

We are pleased to inform you that your manuscript has been judged scientifically suitable for publication and will be formally accepted for publication once it complies with all outstanding technical requirements.

With kind regards,

Eugene A. Permyakov, Ph.D., Dr.Sci.

Academic Editor

PLOS ONE
---

## [Editor Report · Acceptance letter]

10 Feb 2020

PONE-D-19-25528R1 

Investigating the ferric ion binding site of magnetite biomineralisation protein Mms6 

Dear Dr. Staniland:

I am pleased to inform you that your manuscript has been deemed suitable for publication in PLOS ONE. Congratulations! Your manuscript is now with our production department. 

With kind regards,

on behalf of

Prof. Eugene A. Permyakov 

Academic Editor

PLOS ONE